# Antimicrobial Synergistic Effect Between Ag and Zn in Ag-ZnO·*m*SiO_2_ Silicate Composite with High Specific Surface Area

**DOI:** 10.3390/nano9091265

**Published:** 2019-09-05

**Authors:** Jiří Bednář, Ladislav Svoboda, Zuzana Rybková, Richard Dvorský, Kateřina Malachová, Tereza Stachurová, Dalibor Matýsek, Vladimír Foldyna

**Affiliations:** 1Nanotechnology Centre, VSB—Technical University of Ostrava, 17. listopadu 15/2172, 708 33 Ostrava, Czech Republic; 2IT4Innovations National Supercomputing Center, VSB—Technical University of Ostrava, 17. listopadu 15/2172, 708 33 Ostrava, Czech Republic; 3Department of Biology and Ecology, Faculty of Science, University of Ostrava, Dvořákova 7, 701 03 Ostrava, Czech Republic; 4Institute of Geological Engineering, VSB—Technical University of Ostrava, 17. listopadu 15/2172, 708 33 Ostrava, Czech Republic; 5Institute of Geonics of the Czech Academy of Science, Department of Material Disintegration, Studentská 1768, 708 00 Ostrava, Czech Republic

**Keywords:** antimicrobial effect, zinc oxide, silver, synergistic effect

## Abstract

Antimicrobial materials are widely used for inhibition of microorganisms in the environment. It has been established that bacterial growth can be restrained by silver nanoparticles. Combining these with other antimicrobial agents, such as ZnO, may increase the antimicrobial activity and the use of carrier substrate makes the material easier to handle. In the paper, we present an antimicrobial nanocomposite based on silver nanoparticles nucleated in general silicate nanostructure ZnO·*m*SiO_2_. First, we prepared the silicate fine net nanostructure ZnO·*m*SiO_2_ with zinc content up to 30 wt% by precipitation of sodium water glass in zinc acetate solution. Silver nanoparticles were then formed within the material by photoreduction of AgNO_3_ on photoactive ZnO. This resulted into an Ag-ZnO·*m*SiO_2_ composite with silica gel-like morphology and the specific surface area of 250 m^2^/g. The composite, alongside with pure AgNO_3_ and clear ZnO·*m*SiO_2_, were successfully tested for antimicrobial activity on both gram-positive and gram-negative bacterial strains and yeast *Candida albicans*. With respect to the silver content, the minimal inhibition concentration of Ag-ZnO·*m*SiO_2_ was worse than AgNO_3_ only for gram-negative strains. Moreover, we found a positive synergistic antimicrobial effect between Ag and Zn agents. These properties create an efficient and easily applicable antimicrobial material in the form of powder.

## 1. Introduction

Microorganisms play an indispensable positive role in our environment [1,2], but there are situations where they can endanger lives and therefore antibiotics and antifungals have been invented. However, microorganisms acquire resistance to these substances over time and their elimination becomes more difficult [3]. Recently the well-known silver nanoparticles (AgNPs) with their antimicrobial properties became more and more popular again [4], as the silver had a far lower tendency to induce resistance than some conventional antibiotics [5]. The inhibitory effect of silver is based on several processes. It has been shown that silver destabilizes the bacterial membrane and increases its permeability, inactivates respiratory enzymes and proteins responsible for DNA replication, and disrupts ion transport [6,7,8,9,10]. Although silver indiscriminately inhibits protein function, it exhibits limited toxicity to mammalian cells [11]. Silver is broadly used to prevent the growth of microbes on surfaces and within materials like in bacteriostatic and algaecide water filters [12] and in medical fields such as wound dressing materials, implants, breath masks, in sterilization of medical devices and in antimicrobial coatings [13,14,15,16].

AgNPs can be prepared by various methods like chemical [17,18,19] and sonochemical [20] reduction, microwave synthesis [21], radiolysis and photolysis [22] or so called “green” biosynthesis reduction [23,24,25,26]. But in general, wet synthesis from metal salts gives a good control about properties of reduced metal particles while maintains low requirements for laboratory equipment [26,27,28,29].

The antimicrobial activity and usability of silver can be tuned by combining it with other materials in the form of core-shell particles [30,31], or just by putting AgNPs into composite materials with inertia matrix [32,33,34], antimicrobial graphene oxide [35,36,37] or by using magnetic substrate for better subsequent separation [38,39]. The other thing concerning these composites, which should be highlighted, is the possible synergistic effect between two or more antimicrobial agents. This effect was found in Ag-TiO_2_ prepared by solvothermal method [40] and at the similar configuration, the greater specific surface area of the material helps the antimicrobial activity as well [41]. Other studies reported the use of photocatalytic activity of TiO_2_ for enhancing antimicrobial activity [42], or induction of supportive photothermal effect on AgNPs [43]. The positive synergistic effect was also reported for silver with several polymers [44,45,46] and medicaments [47,48] and in combination with other transitional bare metals like Ag [49,50] and Zn, Co, Cd, Ni and Cu [51]. A preparation of AgNPs within silica structure has been reported [52] and also precipitation of AgNPs on ZnO by photocatalysis reduction [53]. The qualitative synergistic effect between these two agents was reported during the wound healing process [54], however the quantitative antimicrobial and even antimicrobial synergistic effect of the combination of Ag with ZnO in silica mesoporous structure has not been properly tested yet. In this paper, we present a preparation of Ag-ZnO silicate composite with high specific surface area and evaluation of its antimicrobial activity against four bacterial strains and one fungus.

## 2. Materials and Methods

The chemicals used were sodium water glass with the modulus = 3 (the molar ration between SiO_2_ and Na_2_O,) from the company Vodní sklo a.s. 110 00 Praha, Czech Republic; zinc acetate dihydrate (Zn(CH_3_COO)_2_·2H_2_O) and silver nitrate (AgNO_3_, 99.8%) were purchased from PENTA s.r.o. 102 00 Praha, Czech Republic. All chemicals used in this study were of analytical grade and were used as received without further purification. For all the preparation of solutions, deionized water was used.

In the synthesis, we used two crucial devices that should be mentioned: (**a**) a self-constructed ultrasonic reactor surrounded by 16 Langevin transducers with maximum acoustic output of 2 kW (power density of 1 kW·dm^−3^) [55] and (**b**) a patented technology of controlled vacuum freeze-drying technique for preparation of non-agglomerated nanostructures [56].

The creation of final antimicrobial material consisted in two steps. In the first step, we prepared the silicate nanostructure substrate by gelation of sodium water glass with m = 3 and a concentration of 15 wt% that was very quickly added to a vigorously stirred (1000 rpm) water solution of zinc acetate that was in stoichiometric excess, and was homogenized by sonification in an ultrasonic reactor for 20 min. In the following precipitation reaction
Na_2_O·*m*SiO_2_ + Zn(CH_3_COO)_2_ → ZnO·*m*SiO_2_ + 2CH_3_COONa(1)
a fine silicate net nanostructure with a zinc content of up to 30 wt% for m = 3 was created, dependent on the used modulus of water glass (Table 1). The morphology of the substrate is like silica gel and the specific surface area of this material reaches values of about 350 m^2^·g^−1^. The resulting silicate net nanostructure was then three-times washed with demineralized water, dispersed in a volume of water at a 1:1 W/W ratio and sonicated under intense stirring of 1000 rpm for 10 min. The dispersion was then rapidly frozen and subjected to vacuum freeze-drying. The residual water was eliminated by heating at 200 °C for 60 min. The resulting material from the first step was zinc oxide silicate ZnO·*m*SiO_2_ with a specific surface area typically above 300 m^2^·g^−1^ with photocatalytic properties that had been already tested and published [57].

The second step was a limited heterogeneous nucleation of Ag within the porous ZnO·*m*SiO_2_ substrate. The dried silicate material ZnO·*m*SiO_2_ was put into the aqueous solution of AgNO_3_ (c = 0.5 mol·dm^−3^) and homogenized by stirring (100 rpm) for 20 min, so that the molecules of AgNO_3_ could sufficiently fill its pores. The solids were then filtered from the dispersion, rapidly frozen and subjected to vacuum freeze-drying. Dried matter was then put into a volume of 200 mL water and under vigorous stirring exposed to 200 nm 10 W UV light irradiation for 100 min. The photocatalytic reduction of AgNO_3_ within limited volumes of the substrate net caused nucleation and growth of Ag nanoparticles that had high surface curvature and low surface energy, which increased the effectiveness of dissociation of silver ions into the environment (Figure 1.).

The material was qualitatively analysed by scanning electron microscopy (SEM) and the energy disperse x-ray spectroscopy (EDX) at SEM FEI Quanta 650 FEG, 627 00 Brno, Czech Republic. The wt% of Ag and Zn were determined by atomic absorption spectrometer with flame atomization (AAS) Unicam 969, 6161 DA GELEEN, The Netherlands. Specific surface area was determined by dynamic analysis of BET isotherm on the device Dynamic BET, the analysis was performed on Qsurf HORIBA SA9601 10200 Praha, Czech Republic. To measure specific surface area—the powder material was degassed for 5 h at 150 °C and then subjected to six-point analysis. X-ray powder diffraction patterns were measured using a Bruker D8 Advance diffractometer (Bruker AXS, 664 84 Brno, Czech Republic). Phase composition of the samples was evaluated using PDF 2 (Release 2011) database (International Centre for Diffraction Data). The optical absorption of the powder solids was obtained by measuring the respective UV-Vis DRS spectra with a Shimadzu UV-2600 (IRS-2600Plus, 190 00 Praha, Czech Republic) spectrophotometer and a spectrometer FLS920 (Edinburgh Instrument Ltd, Kirkton Campus, EH54 7DQ, United Kingdom) was used for registration of photoluminescence spectra.

The bacterial and fungal strains *E. coli* CCM 3988, *Pseudomonas aeruginosa* CCM1960, *Streptococcus salivarius* CCM4046, *Staphylococcus aureus* CCM4223 and *Candida albicans* CCM8186 were obtained from the Czech Collection of Microorganisms (Brno, Czech Republic). Sterile nutrient broth (MPB 10 cm^3^) was inoculated with the bacterial strains and incubated overnight at 37 °C. Yeast *C. albicans* was inoculated in the glucose peptone yeast extract medium (GPY 10 mL) at 28 °C. Microbial suspensions were diluted with a sterile, 0.15 mol·cm^−3^ saline solution to reach a turbidity of McFarland at scale 0.5, which corresponded to concentrations of 1.5 × 10^8^ CFU·cm^−3^. Antimicrobial activities were assessed using the standard dilution micromethod. Disposable microtitration plates were used for the tests. The samples were diluted to concentrations from 1.5 to 53 mg·cm^−3^ with MPB or GPY media. A volume of 100 µL of sample, 80 µL MPB or GPY and 20 µL of suspension of microorganism was added to each well and the microtitration plates were incubated at 37 °C (bacteria) or 28 °C (yeast) at continuous shaking for 24 h. After this incubation period the inoculum was transferred with the help of the inoculate hedgehog to wells with 200 µL of MPB or GPY in new microtitration plates. The minimum inhibitory concentration (MIC) was determined spectrophotometrically at a wavelength of 620 nm after 24 h of incubation at 37 °C or 28 °C. The MIC is defined as the lowest concentration of the tested substance that inhibits the growth of the microbial strain [58,59,60].

## 3. Results

### 3.1. Scanning Electron Microscopy, Energy Disperse X-Ray Spectroscopy and Specific Surface Area Analysis

Backscattered electron images of ZnO·*m*SiO_2_ substrate (Figure 2a) and Ag-ZnO·*m*SiO_2_ (Figure 2b) showed lamellar porous nanostructure for both materials. This morphology is characteristic for the synthesis in the ultrasonic reactor and the subsequent vacuum freeze-drying mentioned in the previous section. The second step of the preparation process filled a part of the substrate’s surface with reduced silver and after the additional mechanical treatment the specific surface area of the material was lowered from 350 m^2^·g^−1^ (ZnO·*m*SiO_2_) to 250 m^2^·g^−1^ (Ag-ZnO·*m*SiO_2_). The deployed silver can be clearly seen as the brighter spots on the material´s surface in Figure 2b.

EDX analysis of the ZnO·*m*SiO_2_ substrate (Figure 3a) showed the expected presence of silicon, oxygen and zinc, while EDX in Figure 3b for Ag-ZnO·*m*SiO_2_ also shows the reduced Ag. The AAS determined the Ag content in the substrate to be 0.27 wt% and the Zn content in both materials reached 28.1 wt%.

### 3.2. Optical Analysis

The UV-Vis absorption spectra of samples are depicted in Figure 4a and the corresponding Tauc plots for optical bandgap estimation in Figure 4b. The optical absorption was affected mainly by photocatalytic ZnO in both samples. The deposition of silver in the second step of the preparation then caused a slight blue shift from 3.25 eV to 3.31 eV in the optical bandgap.

Photoluminescence excitation spectra of both of materials (Figure 5, solid lines) confirmed the existence of optical bandgap under 380 nm. Emission spectra then indicated the blue shift in the UV region for the Ag-ZnO·*m*SiO_2_ and both materials had a high emission in the visible region which related to structural defects in ZnO lattice, namely oxygen vacancies and zinc interstitials. The emission of Ag-ZnO·*m*SiO_2_ in this region was due to the silver shifted more to the longer wavelengths which indicated that silver created other electron states within ZnO and clearly decreased the amount of radiative recombination at higher energies and thus the amount of useful reductive electrons. The optical shifts might have origin at the partial doping of Ag at the diffusion zone in the Schottky junction on the surface of nanocrystalline ZnO.

The XRD analysis in Figure 6 showed the composition of the final Ag-ZnO·*m*SiO_2_ material. All peaks were in good agreement with the powder diffraction file (PDF) of Ag and ZnO (PFD 03-065-3411 and PDF 03-065-2871). The only detected peaks for FCC Ag were (100) and (111) planes due to a low Ag concentration in the specimen. According to XRD analysis, the prepared material was a ZnO with many structural defects (broad peaks), some nanocrystalline ZnO (narrow peaks) and metallic Ag—all in the amorphous silicate SiO_2_ matrix, which obviously did not exhibit any diffraction pattern.

### 3.3. Antimicrobial Activity of Materials

Tests were performed as described in the previous section with five microbial strains at a concentration of the antimicrobial agent of up to 53 mg·cm^−3^. The tested materials were molecular AgNO_3_, ZnO·*m*SiO_2_ substrate and Ag-ZnO·*m*SiO_2_ composite for the antimicrobial testing of Ag, Zn and their combination. We chose pure source of Ag^+^ ions, the AgNO_3_, as a reference instead of colloidal Ag. The concentration equilibrium between Ag nanoparticles and Ag^+^ strongly depends on nanoparticle´s size [61], so we would need equivalent distribution of Ag nanoparticles like we have in our specimen for objective comparison. In the case of AgNO_3_ as a standard, we can compare more materials in the future without the need of dealing with various sizes of Ag nanoparticles.

Results from MIC tests are summarized in Table 2. The smallest MIC for all microbial strains was reached for the AgNO_3_. A higher MIC was detected with the composite Ag-ZnO·*m*SiO_2_ followed by the substrate ZnO·*m*SiO_2_ that represented the antimicrobial activity of zinc in the composite and that did not have MIC in the measurement range for *Candida albicans.* Results above only represent the needed material for antimicrobial inhibition and should not be directly compared without context. We will further provide calculations of the necessary partial MICs for the antimicrobial Ag that will allow us to objectively compare these materials. Furthermore, thanks to the calculated partial MIC, we evaluated also a possible antimicrobial synergistic effect between Ag and Zn. All recalculated MICs are directly proportional to wt% of the antimicrobial agent in the material.

Table 3 shows the values for the calculated partial MIC related to the Ag content using the mean values from Table 2. Those partial MICs showed us the actual concentration of Ag present in the test solution so that we could use them to objectively compare the antimicrobial activity of the composite with respect to the truly needed Ag. The Ag-ZnO·*m*SiO_2_ exhibited the best results for *Streptococcus salivarius* and *Staphylococcus aureus* where the antimicrobial activity was five times better than the pure AgNO_3_. On the other hand, our material was the least effective on *Escherichia coli* and *Pseudomonas aeruginosa* where it showed approximately 2.4 and 3.3 times lower activity than AgNO_3_. Both materials were the least efficient on *Candida albicans* but, in comparison, we needed only 60% of Ag in the case of Ag-ZnO·*m*SiO_2_ composite.

### 3.4. Calculation of Synergistic Effect between Ag and Zn in Ag-ZnO·mSiO_2_ Composite

The positive synergistic effect of two materials is interesting, because in the end we need smaller quantities of both materials in their combination compared to a situation when they are separated. Using the measured MIC data (Table 2) we could determine partial MIC for Ag ($c_{0}^{Ag}$) and Zn ($c_{0}^{Zn}$) by using the mean values of AgNO_3_ and ZnO·*m*SiO_2_ respectively, and partial MICs for Ag ($c^{Ag}$) and Zn ($c^{Zn}$) acting together in the Ag-ZnO·*m*SiO_2_ composite.
(2)$$\mathrm{Synergistic}\;\mathrm{effect}=\frac{m^{Ag+Zn}}{m_{0}^{Ag}+m_{0}^{Zn}}=\frac{2V\left(c^{Ag}+c^{Zn}\right)}{Vc_{0}^{Ag}+Vc_{0}^{Zn}}=\frac{2\left(c^{Ag}+c^{Zn}\right)}{c_{0}^{Ag}+c_{0}^{Zn}}$$

The calculation of the synergistic effect (2) follows this simple thesis that is schematically depicted in the Figure 7: what is the needed effective mass of Ag and Zn ($m^{Ag+Zn}$) in the composite and what is the needed effective mass of separated Ag ($m_{0}^{Ag}$) and Zn ($m_{0}^{Zn}$) in the same volume? Their ratio gives us the synergistic effect and when we subtract it from 1 and multiply by 100, we will get the material savings by using the composite as a percentage. Calculated values are then summarized in Table 4.

Performed calculations of the synergistic effect between Ag and Zn antimicrobial agents in Ag-ZnO·*m*SiO_2_ composite showed material savings of up to 70% for *Pseudomonas Aeruginosa* and around 45% for other strains except for *Candida Albicans*, where the synergistic effect could not be calculated due to the non-existent antimicrobial activity of Zn in this strain within the measured range (Table 2).

## 4. Discussion

Antimicrobial MIC tests summarized in Table 2 showed that all materials acted best on G− *E. coli* bacteria. Another G− multi-resistant bacteria, *Pseudomonas aeruginosa*, was also prone to materials with Ag, but not much for Zn. It was more resilient than the G+ strains. G+ bacterial strains *Streptococcus salivarius* and *Staphylococcus aureus* had almost the same MIC in all tests. The yeast *Candida albicans* made an exception. Despite the fact that the MIC for AgNO_3_ was the same as for the two G+ bacteria, the Ag-ZnO·*m*SiO_2_ composite performed much worse than on all bacterial strains and ZnO·*m*SiO_2_ did not have MIC in the measured range as the Zn content did not even reached the essential inhibitory amount [62].

When we compared the materials with respect to the wt% of the antimicrobial Ag (Table 3), we found out that for G+ strains, the prepared Ag-ZnO·*m*SiO_2_ composite showed five times lower partial MIC than the pure source of Ag^+^ ions, AgNO_3_. This is convenient as both strains are highly resistant to antibiotics [3]. The contrast in efficiency of AgNO_3_ and Ag-ZnO·*m*SiO_2_ for G− and G+ bacteria was probably related to their different cell wall structure. Due to the more complex structure and different composition of the G− bacterial cell wall compared to G+, the nanocomposites were more adsorbed to the surface of G− and deposited in the periplasmic space [63,64]. As a result, the inhibitory effect of Ag from the nanocomposite may be lower than that induced by Ag^+^ ions from AgNO_3_.

The calculations, taking to account partial MIC for Ag and Zn, showed a positive synergistic effect between these two antimicrobial agents with all bacterial strains used. The best result was up to almost 70.5% of the material saving for *Pseudomonas aeruginosa*, the other three had similar savings of 45%. As we used AgNO_3_ as a reference, these values of material savings could be taken as the lowest estimate. *Candida albicans* did not show vulnerability to Zn in the MIC measured range and thus the synergistic effect could not be evaluated.

Our material is not only interesting from the point of view of its antimicrobial activity, but also from the perspective of its preparation. The described technique of light-induced reduction of Ag on the surface of the photocatalyst has already been reported in literature [53]. But we proved that it can be also done with more complex photocatalytic composites—especially those with a high specific surface area, where we needed to nucleate the Ag on the surface of the substrate and where a common chemical reduction would not be suitable. The high photoluminescence of the substrate in the visible region (Figure 5, dashed) meant that photoexcited electrons were trapped in ZnO structural defects and thus the photocatalytic reduction of the Ag acted at a high rate. Due to the fact, that the photocatalytic reaction took place only at the surface of the photocatalyst, the metallic Ag nucleated exclusively within the nanocrystalline ZnO·*m*SiO_2_ substrate. The high specific surface area of 250 m^2^·g^−1^ then eased the solvation of Zn and Ag, which induced an antimicrobial effect in the surrounding area. Although the photoactivity of the material after the addition of Ag was negatively affected, we could still repeat the second step of the preparation procedure, i.e., to additionally expose the material to AgNO_3_ and further increase the amount of Ag in the material.

The Ag-ZnO·*m*SiO_2_ material is an antimicrobial powder that can be easily dispersed by sonification to make possible another specific usage. Due to the fact, that the antimicrobial Ag and Zn are deposited on, respective within the macroscopic and porous silicate structure, it is easy to separate the material back from the dispersion. Moreover, according to our experience, it is feasible to deposit this material onto nanofibers or fabric [65], where it could act as an antimicrobial layer in various types of filters.

## Figures and Tables

**Figure 1 nanomaterials-09-01265-f001:**
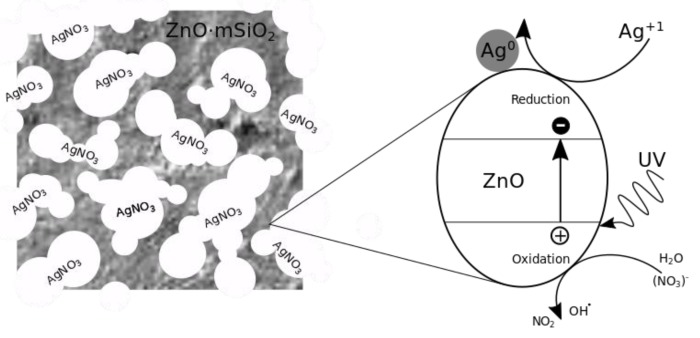
Photoreduction of AgNO_3_ on the surface of photocatalytic ZnO within the pore of the silicate substrate ZnO·*m*SiO_2_. The 200-nm UV light induced an electron-hole charges separation within photocatalytic ZnO. The AgNO_3_ was decomposed and Ag^+1^ reduced to metallic Ag^0^ on the surface of ZnO. The limited amount of AgNO_3_ within pores of the substrate then caused limited growth of Ag particles.

**Figure 2 nanomaterials-09-01265-f002:**
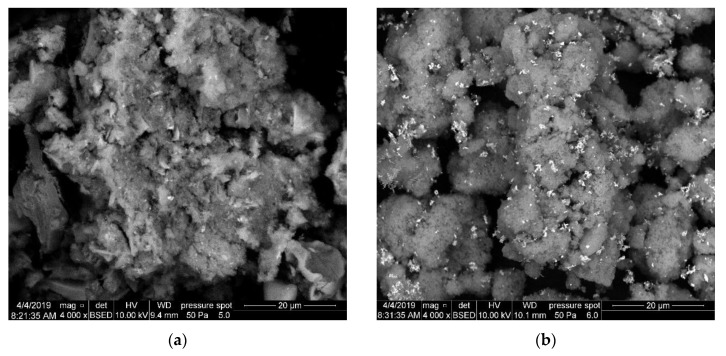
Backscattered electron images of: (**a**) silicate substrate ZnO·*m*SiO_2_ and (**b**) final antimicrobial material Ag-ZnO·*m*SiO_2_.

**Figure 3 nanomaterials-09-01265-f003:**
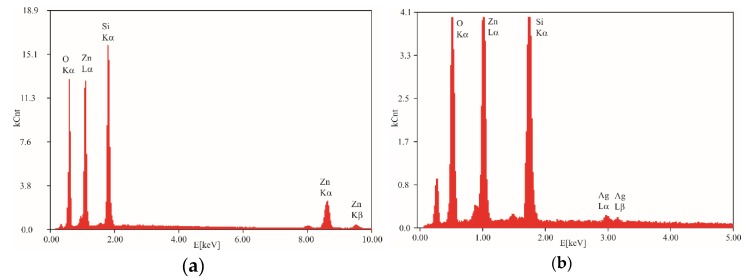
EDX qualitative analysis of: (**a**) silicate substrate ZnO·*m*SiO_2_ and (**b**) final antimicrobial material Ag-ZnO·*m*SiO_2_. Unlabelled peaks have origin in supportive aluminium pad with carbon tape.

**Figure 4 nanomaterials-09-01265-f004:**
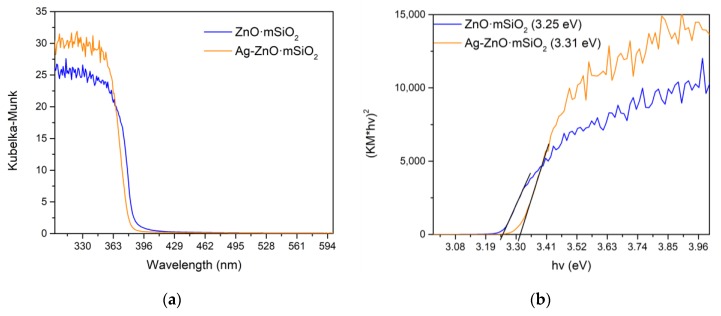
(**a**) UV-Vis absorption spectra and (**b**) Tauc plots with estimated band gap values of Ag-ZnO·*m*SiO_2_ and ZnO·*m*SiO_2_ substrate.

**Figure 5 nanomaterials-09-01265-f005:**
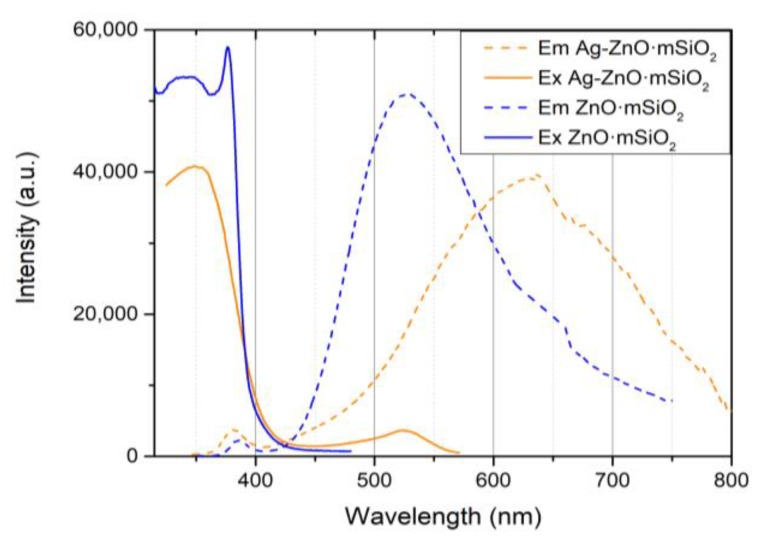
Photoluminescence spectroscopy of antimicrobial material Ag-ZnO·*m*SiO_2_ and ZnO·*m*SiO_2_ substrate. The solid lines are the excitation (Ex) spectra and dashed lines the emission (Em) spectra.

**Figure 6 nanomaterials-09-01265-f006:**
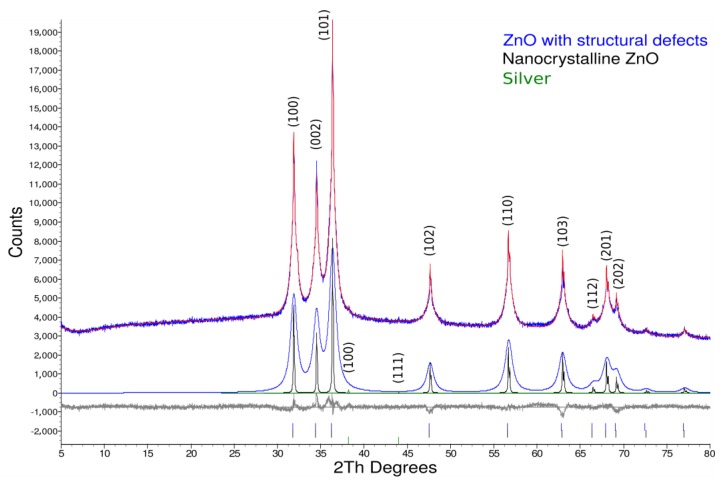
XRD pattern of antimicrobial material Ag-ZnO·*m*SiO_2_. The main top pattern is decomposed to ZnO with structural defects (blue), nanocrystalline ZnO (black) and Ag (green).

**Figure 7 nanomaterials-09-01265-f007:**
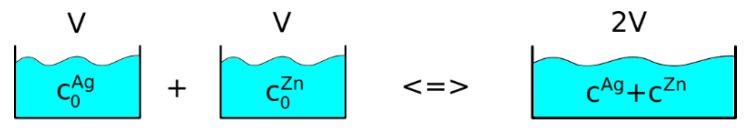
Scheme of the synergistic effect calculation. Was the needed amount of the two separated antimicrobial agents the same as when they acted together?

**Table 1 nanomaterials-09-01265-t001:** Dependence of the silicate composition on the chosen modulus “*m*”. With lower modulus, the zinc content was increasing. The modulus used in the experiment is in bold.

*m* = 3	*m* = 2.5	*m* = 2	*m* = 1.5
ZnO∙3SiO_2_	2ZnO∙5SiO_2_	ZnO∙2SiO_2_	2ZnO∙3SiO_2_

**Table 2 nanomaterials-09-01265-t002:** MIC (mg·cm^−3^) for materials tested on various microbial strains of gram-positive (G+) and gram-negative (G−) bacteria. Values without deviation had the same MIC in all three tests. N/A means that MIC was not found in the measured range.

Microbial Strains	MIC (mg·cm^−3^)		
	AgNO_3_	Ag-ZnO·*m*SiO_2_	ZnO·*m*SiO_2_
*E. coli* (G−)	0.005	2.9 ± 0.1	10.6
*Pseudomonas aeruginosa* (G−)	0.005	3.9 ± 1.4	26.5
*Streptococcus salivarius* (G+)	0.16	5.9 ± 1.3	21.2 ± 2.7
*Staphylococcus aureus* (G+)	0.16	5.9 ± 0.1	21.2
*Candida albicans*	0.16	23.5 ± 0.5	N/A

**Table 3 nanomaterials-09-01265-t003:** Partial MIC of Ag (mg·cm^−3^) with respect to the Ag content in the material and their corresponding concentration ratios. All values have been subsequently rounded.

Microbial Strains	${\mathrm{AgNO}}_{3}\;(c_{0}^{Ag})$	$\mathrm{Ag}-\mathrm{ZnO}\cdot m{\mathrm{SiO}}_{2}\;(c^{Ag})$	Ratio
*E. coli* (G−)	0.003	0.008	2.4
*Pseudomonas aeruginosa* (G−)	0.003	0.010	3.3
*Streptococcus salivarius* (G+)	0.102	0.016	0.2
*Staphylococcus aureus* (G+)	0.102	0.016	0.2
*Candida albicans*	0.102	0.063	0.6

**Table 4 nanomaterials-09-01265-t004:** Calculated partial MIC (mg·cm^−3^) as described in the text and the corresponding synergistic effect in a form of material savings. The *Candida albicans* is not listed, because Zn was inactive in the measured range and the corresponding synergistic effect could not be calculated.

Microbial Strains	$c_{0}^{Ag}$	$c_{0}^{Zn}$	$c^{Ag}$	$c^{Zn}$	Material Saving (%)
*E. coli* (G−)	0.003	2.979	0.008	0.809	45.20
*Pseudomonas aeruginosa* (G−)	0.003	7.447	0.010	1.090	70.45
*Streptococcus salivarius* (G+)	0.102	5.957	0.016	1.652	44.93
*Staphylococcus aureus* (G+)	0.102	5.957	0.016	1.652	44.93

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
