# Peer review of "Antimicrobial Synergistic Effect Between Ag and Zn in Ag-ZnO·mSiO2 Silicate Composite with High Specific Surface Area"

_nanomaterials, 2019, doi:10.3390/nano9091265_

Round 1

Reviewer 1 Report

This paper aims to obtain Ag-ZnO·mSiO2 silicate composite and to verify antimicrobial synergistic effect between Ag and Zn.  The paper don’t reports particular innovation or novelty, but the work is conducted with scientific rigor, clearly written and easy to read. Therefore in my opinion it can be published in the present form. Just the two following suggestions

Introduction

Line 70-71

This sentence concerns the conclusion. Why is it in the introduction? At the end of the introduction, the goal is generally written, or a nod to the experimental design

Results

Line 191

“Results from MIC tests are summarized in Table 2 and visualized in Figure 7”

In my opinion fig 7 can be deleted

Author Response

Dear reviewer,

Thank you for your review.

Line 70-71
I agree the last sentence can be deleted.

And I also deleted the Figure 7 as it does not show anything new.

Reviewer 2 Report

This manuscript describes the antibacterial synergistic effect of Ag-ZnO·mSiO2 silicate composite. The results are interesting and attractive; however, there are some points needed to be revised for further consideration of publication.

1.A wrong spelling is found in the caption of Table 1.

2.The format of Table 1 is not appropriate since there is one blank in the table.

3.UV radiation of 200 nm was applied to reduce AgNO3 to Ag in Ag-ZnO·mSiO2 Silicate Composite. However, the energy density of UV is better to provide in the article.

4.It is better to write CFU in capital instead of cfu since they are abbreviated for colony forming unit.

5.Actually Fig. 2 is a BEI image for presenting the distribution of Ag particles. It is better to write BEI in the caption instead of only scanning electron micrographs.

6.The spectra in Fig. 3 are strongly recommended to re-plot in the same x-axis scale. And a suitable table in the inset is needed, not screenprinting it with some red underlines. Moreover, some peaks without any labels are not suitable.

7.Regarding absorption and photoluminescence spectra, is silver doped in the lattice of ZnO or just absorbed onto ZnO surface? If it is the later as seen in the BEI image, it is not believed to result in the shift of absorption edge as well as emission peaks. The author need to  explain more about this phenomenon.

8.There is no any indication or label to the patterns. How can the readers to know which one belongs to Ag-ZnO·mSiO2?

9.There is no any conclusion in the end. It seems the usual format for this journal.

10.How can the authors obtain actual or correct Ag contents in Ag-ZnO·mSiO2? If this data is from EDS, the results would be not accepted since EDS is not a precise quantisation technique.  

Author Response

Dear reviewer, thank you for your thorough review. I am replying the same way under a numbered comments:

1."A wrong spelling is found in the caption of Table 1."
„contentwas“ - fixed. Thank you.

2."The format of Table 1 is not appropriate since there is one blank in the table."
The table was rearranged.

3."UV radiation of 200 nm was applied to reduce AgNO3 to Ag in Ag-ZnO·mSiO2 Silicate Composite. However, the energy density of UV is better to provide in the article."
I added its power to the text in materials and methods.

4."It is better to write CFU in capital instead of cfu since they are abbreviated for colony forming unit."
It was changed.

5."Actually Fig. 2 is a BEI image for presenting the distribution of Ag particles. It is better to write BEI in the caption instead of only scanning electron micrographs."
It was changed.

6."The spectra in Fig. 3 are strongly recommended to re-plot in the same x-axis scale. And a suitable table in the inset is needed, not screenprinting it with some red underlines. Moreover, some peaks without any labels are not suitable."
I improved spectras. The specimen was on aluminium pad fixed by carbon tape – this is the main source of unlabeled peaks. Some minors are then double and escape energetic lines that have origin in detector. I added a similar text to the figure's description.

7."Regarding absorption and photoluminescence spectra, is silver doped in the lattice of ZnO or just absorbed onto ZnO surface? If it is the later as seen in the BEI image, it is not believed to result in the shift of absorption edge as well as emission peaks. The author need to explain more about this phenomenon."
Our substrate material is highly porous and consists of nanocrystalline ZnO and ZnO with many structural deffects (the XRD). The nanocrystallic character of ZnO most likely caused its partial doping at diffusion zone in Schottky junction on the surface of nanocrystallic ZnO. The similar explanation was also added in the text.

8."There is no any indication or label to the patterns. How can the readers to know which one belongs to Ag-ZnO·mSiO2?"
It is XRD only for Ag-ZnO·mSiO2. The main pattern is divided into ZnO with structural deffects, nanocrystalline ZnO and Ag. It is now more explained in the figure’s description.

9."There is no any conclusion in the end. It seems the usual format for this journal."
Indeed. The template suggested conclusion only if the discussion section is extremely long or complex.

10."How can the authors obtain actual or correct Ag contents in Ag-ZnO·mSiO2? If this data is from EDS, the results would be not accepted since EDS is not a precise quantisation technique."
Thank you for pointing this out. The Ag content was really from EDS and it is wrong. The area for analysis clearly was not sufficiently representative as we measured it again by Atomic absorption spectrometer with flame atomization and results were very different. Although the new values did not change the conclusion about synergistic effect, I had to recalculate values in tables. In the light of the new Ag content, our material is even better than we have thought.

Reviewer 3 Report

Comment:

In this manuscript, the authors synthesized an Ag-ZnO·mSiO2 nanocomposite that combining Ag and ZnO to achieve the synergistic antimicrobial effect. The porous morphology of the final product was observed with SEM. The elemental composition was analyzed and confirmed with EDX, UV-Vis and XRD. In the antimicrobial test, Ag-ZnO·mSiO2 was found to actively inhibit the growth of G+ and G- bacterial strains. By analyzing the MIC data, the authors claim the presence of synergistic antimicrobial effect between Ag and ZnO when applied in the form of Ag-ZnO·mSiO2 nanocomposite. While this is a novel work, there are still some questions that need to be addressed (see comments below). These questions necessitate a minor revision to this manuscript before it could be considered for acceptance.

Additional comments:

The antimicrobial efficacy of Ag-ZnO·mSiO2 was achieved by both nucleated Ag and ZnO. To test the synergistic effect, the control group for Ag-ZnO·mSiO2 should be Ag nanoparticles instead of molecular AgNO3, in which the Ag is in the form of Ag+. Ag nanoparticles might have higher MIC due to the slow Ag+ release. This difference will lead to an inaccurate calculation for the synergistic effect. Please clarify this point. 

Author Response

Dear reviewer, thank you for your review.

It is true that Ag nanoparticles have slower release of Ag+ to the environment. But the concentration equilibrium between Ag nanoparticles and Ag+ strongly depends on nanoparticle´s size. In this case, we would need equivalent distribution of Ag nanoparticles like we have in our specimen for objective comparison. We have chosen the pure source of Ag+ ions, the AgNO3, for this task instead. Then we can compare more materials with the AgNO3 as a standard in the future without the need of dealing with various sizes of Ag nanoparticles.
This indeed affected the precise calculation of synergistic effect. Looking at the values, if we used Ag nanoparticles with expected higher MIC as a reference, we would obtain even better synergistic effect. So our values of material savings could be taken as the lowest estimate. I added similar comment to the manuscript's text.